# AeroPath: An airway segmentation benchmark dataset with challenging pathology and baseline method

**Karen-Helene Støverud**[1], **David Bouget**[1], **André Pedersen**[1,2], **Håkon Olav Leira**[3], **Tore Amundsen**[3], **Thomas Langø**[1,4]*, **Erlend Fagertun Hofstad**[1]

**1** Department of Health Research, SINTEF, Trondheim, Norway, **2** Sopra Steria, Application Solutions, Trondheim, Norway, **3** Department of Thoracic Medicine, St. Olavs Hospital, Trondheim, Norway, **4** Department of Research, St. Olavs Hospital, Trondheim, Norway

* thomas.lango@stolav.no

## Abstract

To improve the prognosis of patients suffering from pulmonary diseases, such as lung cancer, early diagnosis and treatment are crucial. The analysis of CT images is invaluable for diagnosis, whereas high quality segmentation of the airway tree are required for intervention planning and live guidance during bronchoscopy. Recently, the Multi-domain Airway Tree Modeling (ATM'22) challenge released a large dataset, both enabling training of deep-learning based models and bringing substantial improvement of the state-of-the-art for the airway segmentation task. The ATM'22 dataset includes a large group of COVID'19 patients and a range of other lung diseases, however, relatively few patients with severe pathologies affecting the airway tree anatomy was found. In this study, we introduce a new public benchmark dataset (AeroPath), consisting of 27 CT images from patients with pathologies ranging from emphysema to large tumors, with corresponding trachea and bronchi annotations. Second, we present a multiscale fusion design for automatic airway segmentation. Models were trained on the ATM'22 dataset, tested on the AeroPath dataset, and further evaluated against competitive open-source methods. The same performance metrics as used in the ATM'22 challenge were used to benchmark the different considered approaches. Lastly, an open web application is developed, to easily test the proposed model on new data. The results demonstrated that our proposed architecture predicted topologically correct segmentations for all the patients included in the AeroPath dataset. The proposed method is robust and able to handle various anomalies, down to at least the fifth airway generation. In addition, the AeroPath dataset, featuring patients with challenging pathologies, will contribute to development of new state-of-the-art methods. The AeroPath dataset and the web application are made openly available.

**Data Availability Statement:** The AeroPath dataset and the best trained model used in this study are openly available at https://github.com/raidionics/AeroPath, and the ATM'22 dataset can be accessed

at https://atm22.grand-challenge.org. Additionally, the source code for the web application, the AeroPath dataset, and its documentation are publicly available at https://github.com/raidionics/AeroPath. A live demonstration of the web application is also accessible at https://huggingface.co/spaces/andreped/AeroPath.

**Funding:** The author(s) received no specific funding for this work.

**Competing interests:** The authors have declared that no competing interests exist.

## Introduction

Lung cancer is the leading cause of cancer-related deaths worldwide in 2023, accounting for the highest mortality rate among both men and women. The detection of lung cancer at earlier stages, when treatment options are more effective [1], remains a priority for maximizing survival and patient outcome. Most early-stage lung cancer tumors originate from the outer parts of the lung [2]. They are therefore hard to reach by bronchoscopy, the safest and most widely used tool for diagnostic tissue sampling, without a system for planning and live guidance [3]. Anatomically, the airways are first composed of a relatively large and well defined trachea before a divide into right and left main bronchus is reached at the carina of the lungs. Each main bronchus further progresses within each lung, regularly bifurcating and splitting, leading to a complex 3D structure with numerous branches of varying size and direction. For every bifurcation, smaller branches with thinner walls arise, leading to an extremely challenging task of complete identification and segmentation of all airways branches.

Computer tomography (CT) of the chest is the mainstay imaging modality of lung cancer and other lung diseases. A segmentation of the airways from preoperative CT is a prerequisite for image guided diagnostic procedures in the lungs. Bronchial segmentation, commonly known as airway segmentation, has been an attractive research topic for the last decades, with increasing popularity since the release of the public challenge dataset EXACT'09 [4]. The best performing methods from the challenge were mostly revolving around region growing and graph cut algorithms [5], achieving both fast and reliable segmentation of larger central airways but often failing to detect small and thin-walled segments. In more recent years, conventional machine learning-based methods have also been proposed, such as random forest [6], yet not achieving similar performances as the current deep learning-based state-of-the-art methods. However, due to the large dimensions of thoracic CT scans and GPU memory limitations, multi-step or multi-scale designs are commonly used. Garcia *et al.* [7] proposed to use a traditional method for initial segmentation of the bronchial tree, followed by a patch-wise 3D U-Net for refinement of the smaller bronchial branches. Alternatively, a 2D convolutional neural network (CNN) has been applied for the initial segmentation step and a full-volume 3D CNN for further refinement [8]. Single-step designs using efficient 3D CNNs have also been proposed, either through atrous convolutions [9] or using context scale fusion [10]. Instead of heavy 3D architecture, 2.5D variants have shown to be beneficial, using 2D slices from all three orthogonal views as input [11].

Providing guidance under model training or injecting additional anatomical knowledge has been investigated to tackle issues from attempting to segment both large and small airway branches. Traditional algorithmic designs have been suggested, such as using a Free-and-Grow mechanism [12] simulating a deep region growing method, or introducing tubular constraints as regularization during network training [13]. To improve connectivity of the bronchial tree, a graph neural network has been added on top of a U-Net architecture [14]. Alternatively, a step of adversarial refinement [15] or a strategy optimizing integration between small and large branches for a more complete airway tree [16] were also attempted. Finally, more anatomical context can be provided by segmenting a wider range of structures from whole body CT images with a singular model. Chen *et al.* [17] trained a WB-Net in order to segment up to 50 structures, including the lungs and trachea. An even more exhaustive model, based on nnU-Net, was generated to cover 104 structures [18]. The current model can segment both the airway tree and the pulmonary vessels, the latter building upon more traditional image processing techniques for the task [19]. Robust pulmonary vessels segmentation has the potential to be equally valuable during postprocessing given that airways run in parallel with vessels within the lungs to optimize oxygen transfer.

The Multi-site, Multi-domain Airway Tree Modeling (ATM'22) challenge was organized as a part of the MICCAI 2022 conference [20]. The aim was to revolutionize airway segmentation through a larger set of annotated data, task-tailored metrics, and standardized validation protocols. In total, 22 teams successfully participated in the challenge, engendering a leap in airway segmentation performances. The ten best proposed approaches were based of various existing neural network architectures (i.e., WingsNet, Attention U-Net, nnU-Net, 3D U-Net, or 3D ResNet) either through specific modifications or overall combinations and ensembling. Zhang *et al.* [20] identified three common strategies among the participants that led to higher quality segmentation. First, multi-step designs, performing first a lung masking step, have shown to prevent leakages which resulted in a more complete airway segmentation. Second, additional information extracted from the data, such as centerlines, radius, or location of the branches, may be used to improve intra-class discrimination. Finally, the loss function must be designed to capture the differences between airway branches at multiple scales (i.e., large, intermediate, and small scales). Two examples successfully taking topological completeness into account were the general union loss and the Jaccard continuity and accumulation mapping loss.

To validate and benchmark newly proposed segmentation methods, such public challenge datasets, with determined training, validation, and test subsets, are essential. However, the EXACT'09 dataset, the BAS dataset [21], and several other datasets (cf. Zhang *et al.* (2023) [20]), share the same limitations. All consist of less than 100 samples which is slightly insufficient to train robust deep learning models. To address this drawback, ATM'22 provided 500 annotated non-contrast thoracic CT scans, including both screening and COVID-19 patients data. While the dataset magnitude is superior to other existing ones, few patients with severe pathology were included, and as such does not represent a suitable benchmark dataset for airway segmentation in CT scans from patients with lung tumors. A reliable method for such patients, where the airway tree appearance can be moderately to greatly impacted by the tumors, is required to be integrated in systems for planning and guidance of bronchoscopy procedures.

In this paper, our first contribution is the release of a new public benchmark dataset called AeroPath, comprising of 27 annotated contrast CT scans with severe pathologies including tumors of various size. As second contribution, a multi-scale fusion design for automatic airway segmentation was designed, trained on the ATM'22 dataset, and evaluated on our benchmark dataset. Finally, an open tool for airway segmentation was developed, validated against other existing open software solutions.

## Materials and methods

### Ethics statement

The collection of the CT data used in this study was performed in accordance with an institutional review board approval from the Norwegian regional ethics committee (REK Sør-Øst A) in Oslo, Norway, reference 2010/3385a, and accessed for research purposes from first approval date of March 16, 2011 to December 2023. Written informed consent was obtained from patients included. All methods were carried out in accordance with relevant guidelines and regulations.

### AeroPath benchmark dataset

For this study, the AeroPath dataset, comprising of 27 computed tomography angiography (CTA) scans acquired using the Thorax Lung protocol at St. Olavs hospital (Trondheim,

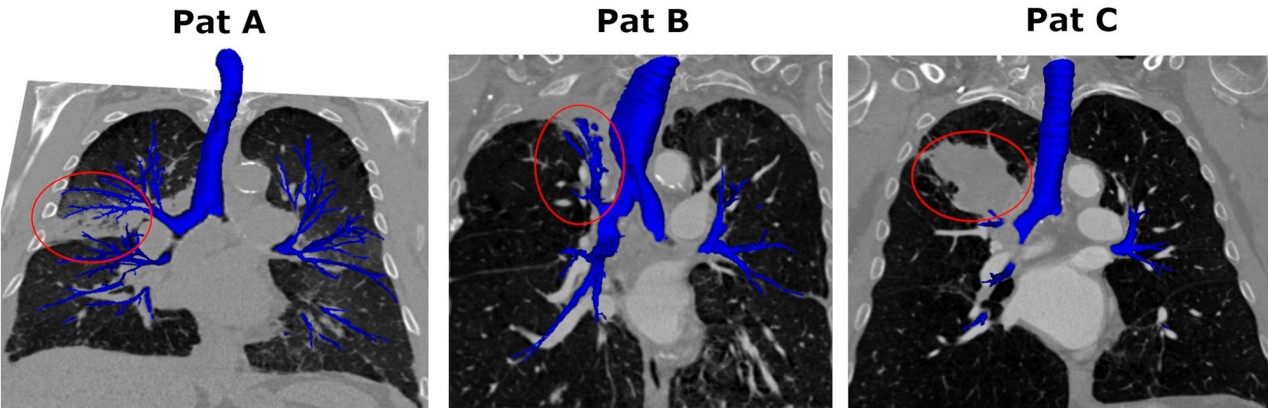

**Fig 1. Example coronal slice of CT patients featured in the AeroPath dataset, with 3D ground truth (blue) and pathology highlighted (red circle).** Patient A has signs of an infection, patient B has dilated trachea and a collapsed segment in the upper left lobe, and patient C has a large tumor in the upper left lobe.

Norway), was assembled. The AeroPath dataset is made openly available at https://github.com/raidionics/AeroPath.

The included patients (nine women), aged 52 to 84 (median 70), were all undergoing diagnostic tests for lung cancer and had a wide range of pathologies including malignant tumors, sarcoidosis, and emphysema, three examples from the dataset are shown in Fig 1. Overall, the CT scan dimensions are covering $[487: 512] \times [441: 512] \times [241: 829]$ voxels, and the transaxial voxel size ranges $[0.68: 0.76] \times [0.68: 0.75]$mm$^2$ with a reconstructed slice thickness of $[0.5: 1.25]$ mm. The annotation process for generating the ground truth was performed in three steps. First, the largest components (i.e., trachea and the first branches) were extracted based on a region growing [22, 23] or a grow-cut method [24]. Due to leakage, the region growing method did not yield satisfactory results in all cases. Therefore, for certain cases, the grow-cut method in 3D Slicer [25] was used instead. In the second step, BronchiNet [7] was employed to segment the smaller peripheral airways. In the third and final step, the segmentations were refined manually. Bronchial fragments and missed segments were connected, before false positives and fragments that could not be connected based on visual inspection were removed. All manual corrections were performed using the default segment editor in 3D Slicer. The manual correction was performed by a trained engineer, supervised by a pulmonologist. Finally, all annotations were verified on a case-by-case basis by a pulmonologist. The final annotations from the AeroPath segmentation included on average 128±56 branches per CT scan (see Table 1).

**Table 1. Overview of the two datasets used in this study (i.e., ATM'22 training data and AeroPath).** In addition, an overview of the validation and test set used in the ATM'22 challenge is provided.

| Test set | | # CTs | # of annotated branches |
|---|---|---|---|
| ATM'22 | training data | 300 | 208±74 |
| ATM'22 | validation data | 50 | 212±52 |
| ATM'22 | test data | 150 | 179±49 |
| ATM'22 | test, COVID-19 only | 58 | 167±17 |
| AeroPath | | 27 | 128±56 |

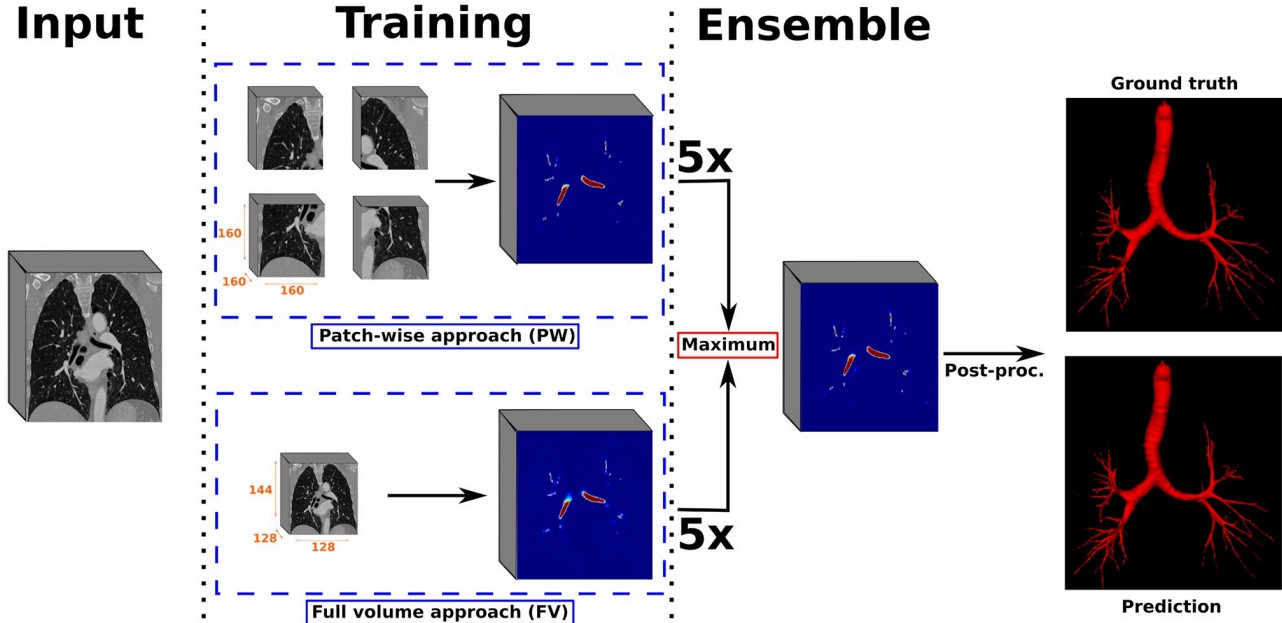

**Fig 2. Overall airways segmentation approach over contrast-enhanced CT scans.** Models were trained using the AGU-Net architecture, both in a full volume and patch-wise fashion under a 5-fold cross-validation paradigm, before being ensembled. Finally, a postprocessing step was performed for refining the binary predictions.

### Airway tree segmentation approach

For the airway segmentation task, the selected backbone architecture is the Attention-gated U-Net (AGU-Net) [26], which has shown to perform well for the segmentation of other anatomical structures in both MRI and CT scans [27, 28]. We also considered alternative backbones, including the no-new UNet (nn-UNet). However, the primary strength of nn-UNet lies in its preprocessing steps, a feature we have seamlessly integrated into our training framework. The overall pipeline illustrating the different steps of our proposed method is presented in Fig 2.

**Full volume and patch-wise training.** To prepare the training samples, all CT volumes were preprocessed identically using the following series of steps: (i) resampling to an isotropic spacing of 0.75 mm$^3$ using spline interpolation of order one from NiBabel, (ii) lung-cropping using a pre-trained network [29] in order to generate the tightest bounding box around the lungs, and (iii) intensity clipping to the range [−1024, 1024] Hounsfield units followed by normalizing the intensities to the range [0, 1]. For the full volume approach, all CT volumes were downsampled to a resolution of $128 \times 128 \times 144$ voxels. For the patch-wise approach, all CT volumes were split into non-overlapping patches of $160 \times 160 \times 160$ voxels.

As architecture, the single-stage AGU-Net approach leveraging multi-scale input and deep supervision to preserve details, coupled with a single attention module, was employed. The final architecture had five levels and {16, 32, 128, 256, 256} as convolution blocks, without batch normalization layers. The AGU-Net architecture used is depicted in details in Fig 3.

For training, the loss function used was the class-averaged Dice loss, excluding the background. For the patch-wise strategy, training has been performed with mixed precision, batch size four, and using the Adam optimizer with an initial learning rate of 0.0005. In addition, a gradient accumulation with eight steps was performed, resulting in a virtual batch size of 32 samples [30]. The number of updates per epoch has been limited to 512, and an early stopping

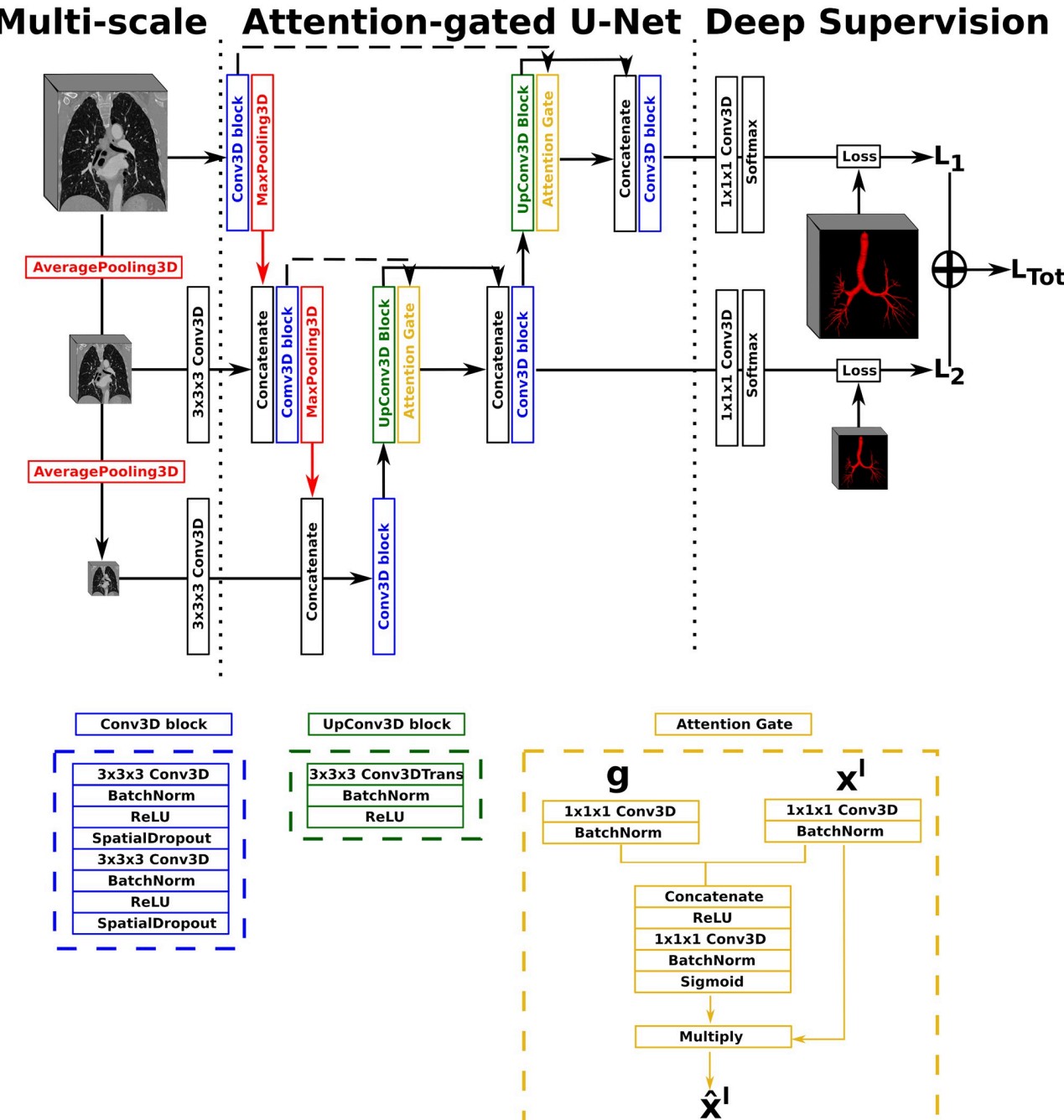

**Fig 3. Illustration of the Attention-Gated U-Net (AGU-Net) architecture design, with multi-scale input and deep supervision.** The representation only shows three levels for viewing purposes while five levels have been used in this work.

scheme was setup to stop training after 15 consecutive epochs without validation loss improvement. For the full volume approach, training has been performed with full precision, the same real and virtual batch sizes, and an initial learning rate of 0.001.

For the data augmentation strategy, the following transforms were considered: horizontal and vertical flipping, random rotation in the range $[-20, 20]°$, translation up to 20% of the axis

dimension, zoom in the range [50, 150]%, and random gamma correction in the range [0.5, 2.0]. For each input sample during training, each data augmentation strategy was considered independently and applied consecutively with a probability of 50%. A random value was then sampled in the aforementioned ranges for the data augmentation methods to be applied.

## Model ensembling and postprocessing

Multiple models operating on different input shapes or focusing on different aspects during training can be ensembled to generate better prediction probability maps [31]. Training on full CT scans using their original dimensions would yield the best segmentation performances, as both global context and local details could be simultaneously leveraged. However, due to hardware constraints in terms of available memory, training in such configuration is unachievable. As a results, the ensembling of full volume models at a lower resolution and patch-wise models represents the best compromise for generating predictions consistent along the trachea and main bronchi, while remaining precise enough along the smaller bronchus deep inside the lungs. The ensemble strategy followed in this study was through the use of a voxel-wise maximum operator applied over the multiple probability maps.

After thresholding the ensembled probability map, the binary airways mask was further refined by a postprocessing algorithm to connect free segments and filter false positives. The centerlines of all segmented parts were extracted, and the main airway tree was identified. At all endpoints of the airway tree, a search was performed to close free segments with similar orientation to the end of the airway tree. Noisy free segments were excluded by limiting the accepted orientation variance between the centerline points. After the centerline segments were connected, artificial tubes matching the airway radius were created around the added part, connecting the main airway tree with the free segments in the binary airway mask. Finally, all remaining free segments still not connected to the main airway tree were discarded.

## Validation studies

**Data.** In this work, two datasets were leveraged to train and test the different methods considered. First, our newly introduced AeroPath dataset. Second, the Airway Tree Modeling Challenge dataset from 2022 (ATM'22) [20], including 300 large-scale CT scans with detailed pulmonary airway annotation. The dataset is public, was collected from different sites, and with diverse health conditions of the scanned subjects ranging from healthy subjects to COVID'19 patients and patients with severe pulmonary disease. The ATM'22 dataset was used for both training and testing of the methods, while the AeroPath dataset was solely used for testing. Although AeroPath is relatively small in size, it should be noted that the ATM'22 dataset contains images both from the EXACT'09 and LIDC-IDRI datasets (20 and 54 CT images included, respectively). Looking ahead, the AeroPath dataset will be expanded, and it may even be incorporated into larger datasets.

**Protocols.** All the models trained with the AGU-Net architecture followed the same 5-fold cross-validation paradigm over the ATM'22 dataset, whereby at each training iteration one fold was used as validation set, one fold as test set, and the remaining folds constituted the training set. All the baseline methods or models were already pretrained and direct inference was performed to compute performance metrics. All methods and models were then applied on the AeroPath dataset for unbiased benchmarking.

**Baseline methods.** Multiple external baseline methods were used to properly benchmark our proposed final method. First, the segmentation method for tubular structures implemented in FAST [32], relying on centerline extraction and parallelized region growing, was included (noted FAST). Second, FAST results were fused with BronchiNet [7], an end-to-end

optimised airway segmentation method based on the U-Net architecture (noted FAST + BronchiNet). Third, the TotalSegmentator model [18] was considered, trained to segment 104 anatomical structures, including the trachea and peripheral bronchus, from a total of 1024 CT examinations using the nnU-Net framework (noted TotalSegmentator). Finally, the MED-Seg model [33] was included, contender for the ATM'22 challenge, using a 2.5D modified EfficientDet architecture [34, 35] (noted MEDSeg). Unfortunately, none of the five best performing methods from the ATM'22 challenge could be run, by lack of code or trained model availability.

When using the AGU-Net architecture, seven different variants were considered: i) using the entire downsampled CT volume as input to the network (noted FV), ii) a 3D patch-wise approach extracting patches from isotropic resolution (noted PW), iii) ensemble of five folds models from i) (noted FV ensemble), iv) ensembling of the five folds models from ii) (noted PW ensemble), v) ensemble i) and ii) (noted FV-PW), vi) ensemble iii) and iv) (noted FV-PW ensemble), and vii) same as vi) but adding the postprocessing refinement method (noted FV-PW ensemble + postproc.).

**Metrics.** The metrics used by the ATM'22 challenge were used to compare methods in this study, computed in the exact same fashion following the official implementation [36]. Common voxel-wise segmentation task metrics such as the Dice similarity coefficient (DSC), precision, sensitivity, and specificity are included;

$$DSC = \frac{2|X \cap Y|}{|X + Y|}, \tag{1}$$

where X is the predicted airway tree and Y the ground truth. (Note that the absolute signs means a sum over all voxels.) Next we define precision and sensitivity, which measures the voxelwise accuracy of the segmentation:

$$\text{Precision} = \frac{|X \cap Y|}{|X|}, \tag{2}$$

$$\text{Sensitivity} = \frac{|X \cap Y|}{|Y|}. \tag{3}$$

The specificity measures how many voxels that are marked as true negative, this can be given as

$$\text{Specificity} = \frac{|I| - |X \cup Y|}{|I| - |Y|}, \tag{4}$$

where $I$ represents the entire image.

In addition, specific property-related metrics are also covered to assess topological completeness, such as the tree length detected rate (TD) and branch detected rate (BD). As a preprocessing step for computing the metrics, the airway tree must be skeletonized and divided into branches, which was also performed using the aforementioned official implementation [36]. Based on the skeletonization the total length of all detected branches are found in the ground truth ($T_{\text{ref}}$) and predictions ($T_{\text{det}}$) TD is calculated

$$TD = \frac{T_{\text{det}}}{T_{\text{ref}}} \tag{5}$$

The branch detection rate is the ratio between the number of correctly predicted branches ($B_{det}$) and the total number of branches in the ground truth ($B_{ref}$), i.e.,

$$BD = \frac{B_{det}}{B_{ref}}. \tag{6}$$

Since the AGU-Net models were trained on CT scans clipped around the lungs, and to ensure fair comparison between all the considered methods, all ground truth and automatic segmentation results had the trachea clipped automatically at the top of the lungs.

## Results

Experiments were conducted on multiple machines with the following specifications: Intel Xeon W-2135 CPU @3.70 GHz CPU, 64 GB of RAM, NVIDIA Tesla V100S (32GB VRAM) dedicated GPU, and a regular hard-drive. The implementation was done in Python 3.7, using TensorFlow [37] v2.8 for training the segmentation models, and ONNX Runtime [38] v1.12.1 for performing model inference. A web application was developed using Gradio [39] v3.50.2 for others to easily test the proposed model on their own data. A live demonstration is also made available at Hugging Face Spaces. The web application and its source code can be accessed at https://github.com/raidionics/AeroPath.

### Evaluation on ATM'22 the dataset

In Table 2, the performance metrics from the PW ensemble method over the ATM'22 training dataset are detailed. The decision to report results for this method was driven by the trade-off between runtime and segmentation performances. The average DSC reached 89.91±5.54% and was relatively stable across the five folds. The TD of 79.91±10.10 is acceptable, whereas a BD of 64.88±14.16 is relatively low indicating a high number of false negatives. As shown in Fig 4, example (a) is of high quality apart from the last generation of branches in the ground truth. Example (b) achieves high DSC (95%), TL (90%), and BD (81%), whereas example (a) achieved lower scores, especially for BD. The proposed method performed well on precision, sensitivity, and specificity.

### Evaluation on the AeroPath dataset

Performance metrics for five variants of the AGU-Net design have been reported in Table 3. The FV ensemble model, using downsampled full volumes, achieved similar scores compared to the other methods for all metrics apart from TD and BD. The downsampling causes loss of details and fine structures are not captured (see Fig 5). The patch-wise segmentation methods (PW), with and without ensembling, perform better than the full volume for all metrics. The PW method has the best overall DSC (84.98 ± 3.24) of all tested methods. Adding ensemble to

**Table 2. Airway segmentation performance over the ATM'22 training dataset using the best-performing configuration of the AGU-Net architecture (PW ensemble).** TD and BD are abbreviations of detected tree length and branch detection rate, respectively.

| Fold | TD (%) | BD (%) | DSC (%) | Precision (%) | Sensitivity (%) | Specificity (%) |
|------|--------|--------|---------|---------------|-----------------|-----------------|
| 0 | 71.38 ± 05.53 | 53.97 ± 10.04 | 88.24 ± 04.06 | 93.65 ± 05.71 | 99.99 ± 00.01 | 83.81 ± 05.57 |
| 1 | 83.93 ± 07.27 | 70.34 ± 12.68 | 90.43 ± 02.56 | 93.71 ± 03.86 | 99.99 ± 00.01 | 87.67 ± 04.93 |
| 2 | 79.67 ± 13.15 | 64.67 ± 15.80 | 89.80 ± 10.36 | 94.11 ± 10.62 | 99.99 ± 00.01 | 86.10 ± 11.02 |
| 3 | 83.78 ± 07.64 | 69.85 ± 12.89 | 90.72 ± 03.59 | 94.99 ± 03.03 | 99.99 ± 00.01 | 87.13 ± 06.37 |
| 4 | 80.81 ± 07.88 | 65.59 ± 12.74 | 90.36 ± 02.89 | 92.77 ± 02.72 | 99.99 ± 00.01 | 88.27 ± 05.22 |
| Total | 79.91 ± 10.18 | 64.88 ± 14.16 | 89.91 ± 05.54 | 93.85 ± 05.95 | 99.99 ± 00.01 | 86.60 ± 07.12 |

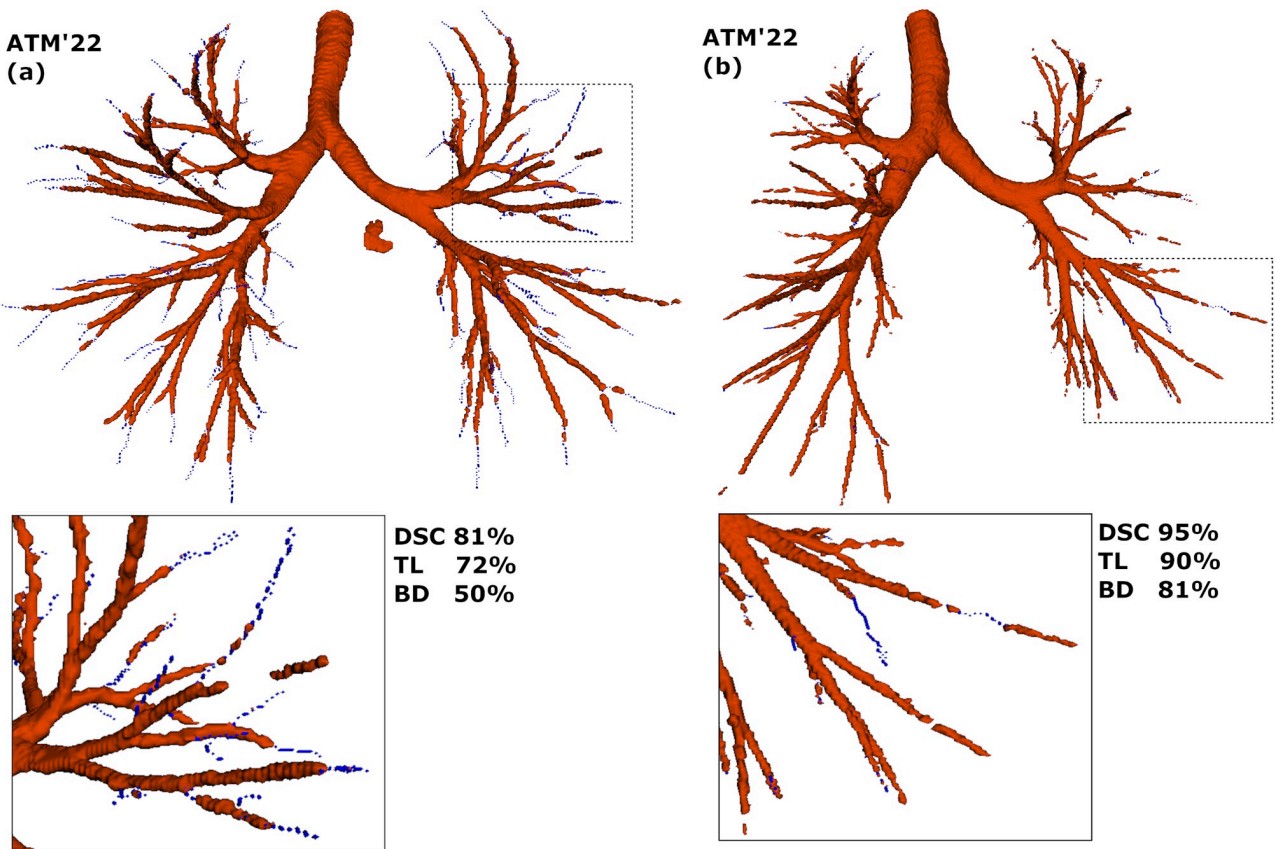

**Fig 4. Two example cases from the ATM'22 dataset with the predictions in red and annotation in blue.** (a) The predictions match the annotation closely apart from the last generation of branches. As a consequence, TL and especially the number of detected branches and BD is low. In (b) the predictions and annotations are very similar and the TL and BD are substantially higher.

the PW model causes a drop in DSC and precision with 0.5 and 4%, respectively. On the other hand, TD and BD increases with 2.4 and 4.1%, respectively. Combining FV and PW lead to a further decrease in DSC and precision, and an increase in BD and specificity, whereas TD only increased slightly (0.2%). Finally, adding postprocessing does not have a substantial impact on any of the evaluation metrics. However, the added value of the postprocessing step to improve

**Table 3. Airway segmentation voxel-wise performances from the developed methods and considered baselines, over the AeroPath dataset.** The first five methods are all based on the AGU-Net architecture, and the next four are open-source baselines. FV stands for full volume, PW for patch-wise, TD for detected tree length, and BD for branch detection rate. †Note that the ground truth annotations were partly based on FAST + BronchiNet.

| Methods | TD (%) | BD (%) | DSC (%) | Precision (%) | Sensitivity (%) | Specificity (%) |
|---|---|---|---|---|---|---|
| FV ensemble | 48.87 ± 11.21 | 34.94 ± 07.59 | 83.88 ± 03.51 | 89.18 ± 08.50 | 99.99 ± 00.01 | 80.13 ± 06.91 |
| PW | 91.80 ± 03.50 | 84.67 ± 07.11 | 84.98 ± 03.24 | 87.23 ± 06.56 | 99.98 ± 00.01 | 83.87 ± 09.19 |
| PW ensemble | 94.18 ± 02.57 | 88.75 ± 05.25 | 84.21 ± 02.80 | 83.22 ± 07.55 | 99.98 ± 00.01 | 86.44 ± 07.95 |
| FV-PW ensemble | 94.28 ± 02.58 | 89.17 ± 05.35 | 83.75 ± 03.19 | 80.91 ± 08.46 | 99.97 ± 00.01 | 88.07 ± 06.95 |
| FV-PW ensemble + postproc. | 94.06 ± 03.30 | 89.38 ± 05.79 | 83.13 ± 03.69 | 79.95 ± 08.95 | 99.97 ± 00.01 | 87.95 ± 07.00 |
| FAST [23] | 48.76 ± 13.69 | 37.33 ± 11.67 | 80.38 ± 16.94 | 94.91 ± 19.53 | 99.89 ± 00.53 | 72.47 ± 10.82 |
| FAST + BronchiNet [7]† | 85.16 ± 06.67 | 74.71 ± 10.91 | 84.36 ± 18.06 | 93.75 ± 19.16 | 99.89 ± 00.53 | 79.91 ± 13.02 |
| TotalSegmentator [18] | 74.79 ± 10.43 | 58.88 ± 13.94 | 82.27 ± 03.27 | 83.22 ± 08.66 | 99.98 ± 00.01 | 82.53 ± 06.86 |
| MEDSeg [33] | 73.87 ± 13.11 | 65.90 ± 15.31 | 84.90 ± 04.06 | 90.82 ± 06.78 | 99.99 ± 00.01 | 80.98 ± 10.01 |

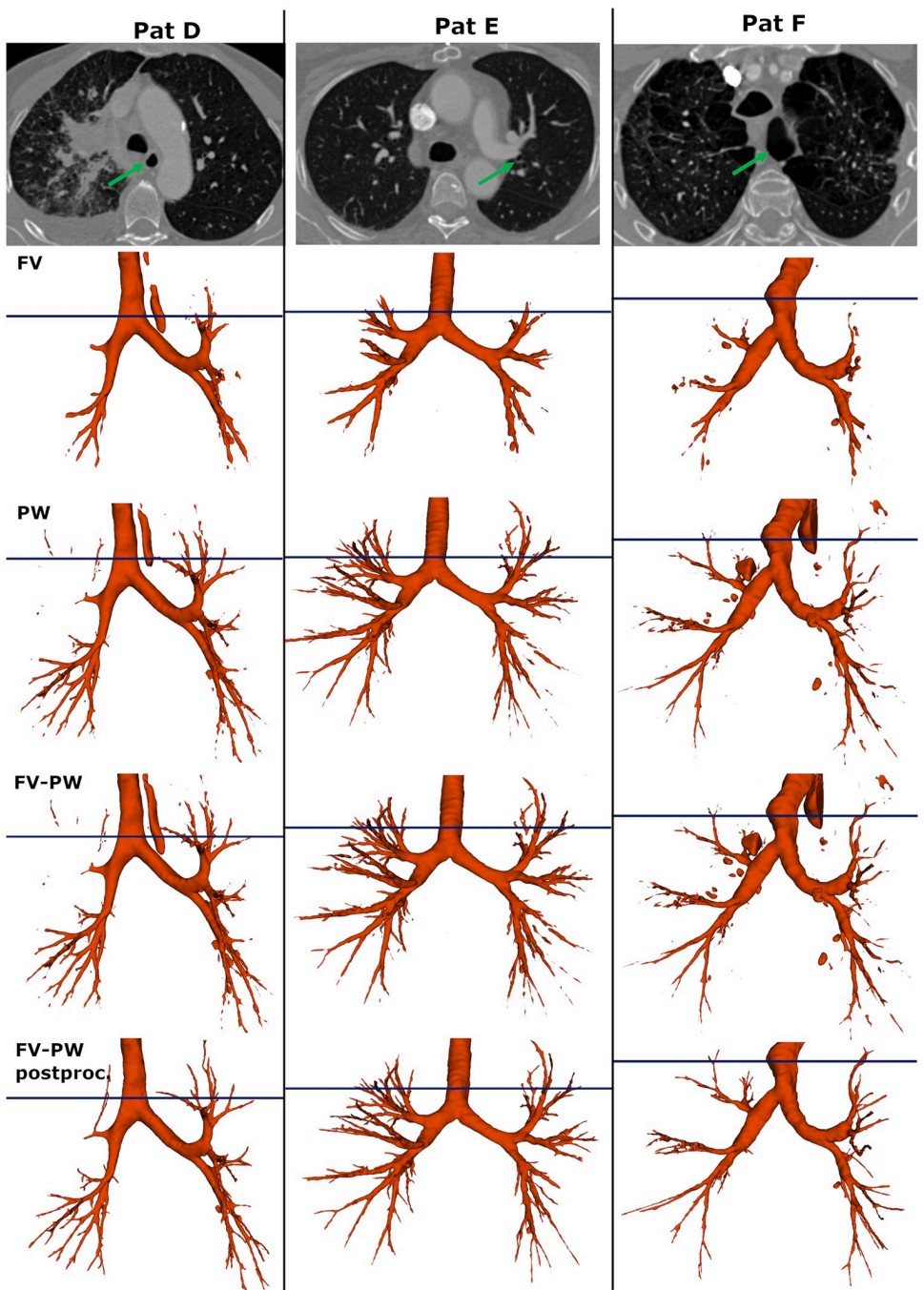

**Fig 5. Comparison of predictions based on the tested AGU-Net models with ensemble; FV, PW, FV-PW, and FV-PW postproc.** The top row displays an axial slice from the CT images of the example cases D, E, and F. The green arrows points out the large esophagus in D, large and distinct airways in E, and the abnormal lung segment in F. The horizontal lines show the level of the axial slices. The FV method successfully segments the main branches, but not the small ones. Even though FV produces relatively few false positives the esophagus is included in D. The ensembled PW method has the overall best metrics. Note the many false positive in F, where parts of the upper right lobe are misinterpreted as an airway. Combining FV and PW leads to a more complete segmentation, but also yields increased false positives. The importance of postprocessing is evident from the last row, both removing islands and connecting fragmented segments.

the final segmentation can be visually ascertained (see Fig 5). Fragmented branches are connected back together, and in addition branches still unconnected and islands are removed.

Compared to the baseline methods, all AGU-Net designs are better on tree length detection rate and branch detection rate. However, the MEDSeg algorithm is best on DSC. For all designs, the specificity is relatively low, indicating a higher number of false positive segments. Performance metrics have not been reported in the table for BronchiNet alone, since exclusively small airway branches were generated from it. Only its combination with FAST, used for segmenting the trachea and the major branches, is reported. In addition, the ground truth was based on a combination of BronchiNet and segmentation, performed in FAST or 3D Slicer. As such, the mean DSC and other metrics are expected to be relatively higher. However, the standard deviation is higher than for the other algorithms, which is expected since BronchiNet and FAST perform poorly on some of the cases featured in the AeroPath dataset. The results based on FAST only, more potent for segmenting the major airway branches, show that a relatively high DSC score (83.84%) can still be achieved in spite of poor segmentation performances over the smaller airway branches. A TD of 49.39% and a BD of 37.27% demonstrates that FAST failed to segment smaller branches.

The predictions from three different patients are provided in Fig 6, comparing the AGU-Net design with postprocessing, MedSeg, and TotalSegmentator methods. While patient A has signs of infection (see Fig 1), the ground truth covers up the sixth generation of branches. Even if the DSC metric is similar for all cases, only the AGU-Net predictions reach a TD of 91% for this specific case, whereas TotalSegmentator and MedSeg achieve a TD of 74 and 58%, respectively. Patient B exhibits a wider trachea than usual and a collapsed lung segment is visible in the upper left lobe. It is therefore a challenge for the models to segment the airways of Patient B correctly due to both these abnormal characteristics. After the inclusion of zoom augmentation during training of the AGU-Net based models, successful segmentation of the trachea in all patients featured in the AeroPath dataset was achieved. The AGU-Net model also managed to segment many of the abnormal segments in the collapsed area in Patient B. TotalSegmentator also has a complete segmentation of the trachea even for patient B and manages to segment out most of the difficult areas, ending up with a TD of 75%. Indeed, TotalSegmentator has been trained for specifically being able to segment the trachea as one category, and the complete airway tree as another category, making it more robust to such challenges. Unfortunately, the MedSeg model fails both on the wide trachea and the collapsed segment, which is reflected in both a low DSC of 75% and a low TD of 56%. Regarding the last example (i.e., Patient C), a large tumor is featured in the upper left lobe and the ground truth covers only a limited amount of branches as a result. The TD is hence high for all three models and the AGU-Net model has surprisingly correctly identified branches that are not present in the ground truth. Nonetheless, segmented branches missing from the ground truth are considered as erroneous and marked as false positives, which resulted in a lower DSC than expected.

## Discussion

In this study, we presented a novel dataset consisting of 27 contrast-enhanced CT scans with corresponding airway tree annotations, AeroPath. All included patients suffer from major lung-related pathology, representing a wide range of challenges for automatic segmentation methods. In order to alleviate the burden of airway segmentation for patients extensively affected by pulmonary pathology, we proposed an open tool using the AGU-Net architecture. Both a full volume and a patch-wise approach were investigated separately, before ensembling was implemented to enable the detection of a higher number of airway branches. The

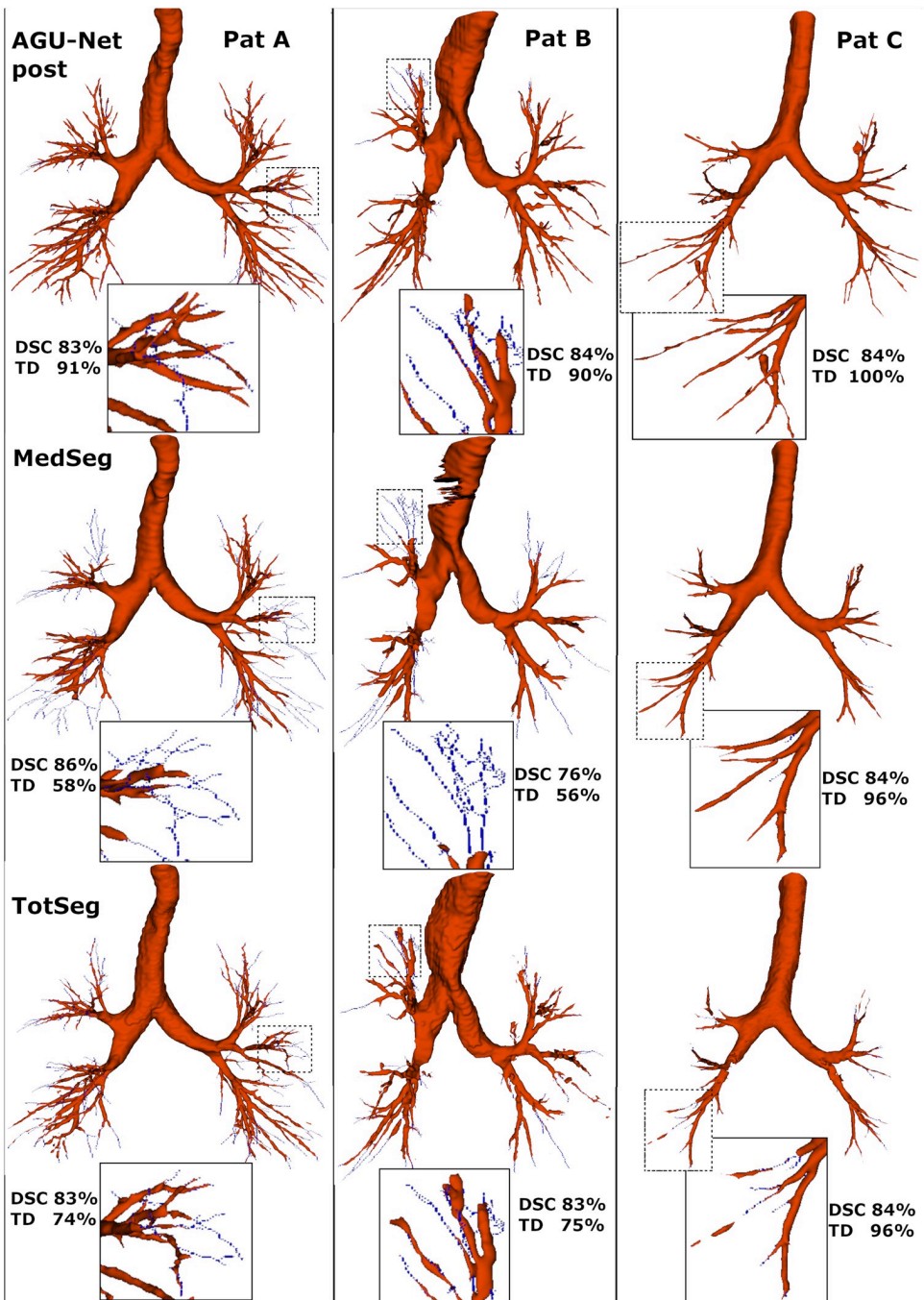

**Fig 6. Segmentation results comparison between the AGU-Net design including ensemble and postprocessing, and two open-source baselines: MedSeg and TotalSegmentator.**

inclusion of task-related data augmentation methods (e.g., zoom) during training was shown to be paramount for a more robust segmentation of the trachea and branches at various scales, especially under heavy anatomical pathology-induced deformations (e.g., enlarged trachea, tumors, and emphysema). The aim of the study was to create clinically valuable airway tree masks for use in planning and guidance of bronchoscopy. To that extent, a tailored

postprocessing step was further supplemented for connecting isolated fragments and removing false positives. The proposed method has been integrated into the open solution Fraxinus [40], research software for planning and guidance of bronchoscopy, in use within several hospitals in Norway.

The best performing AGU-Net variant predicted highly complete segmentations for all patients included in the AeroPath dataset, with high scores on the metrics describing topological completeness. The results also showed that the proposed AGU-Net postproc. method performs better than the region growing method and readily available trained models (e.g., TotalSegmentator and MEDSeg). As mentioned in the results and the caption of Table 3, the FAST + BronchiNet method has been used to initially generate the ground truth segmentations over AeroPath patients. However, an extensive manual work has been additionally performed in 3D Slicer to refine the initial segmentations into proper and more complete ground truths, including many omitted smaller branches and leakages in the region growing method. Therefore, the impact of a potential bias in the computed metrics over the FAST + BronchiNet baseline has been drastically minimized by the extent of manual refinement. Segmentation performances for the ATM'22 were also provided, but solely over the training subset from the challenge. Access to the validation or test subsets was impossible, and only a participation in the challenge could have provided us with the correct metrics. Hence, the computed metrics reported in the current study are not directly comparable to the challenge results [20], but some extent of insight can be inferred. The AGU-Net based models performed better on the ATM'22 dataset than on AeroPath with respect to DSC and precision. However, the topological completeness metrics dropped substantially for the ATM'22 dataset. On average, AGU-Net models failed to detect the smallest branches deep inside the lungs, mandating for a better handling of smaller branches for example through the introduction of a tailored loss function (e.g., general union loss [41]). On the other hand, our proposed approach aims to put more emphasis on correct segmentation of the airway tree in patients with heavy anatomical deformations and as such focuses less on smaller branches, often not directly observable nor reachable with a bronchoscope. The MEDSeg architecture, a contender in the ATM'22 challenge, achieved similar metric performances on the ATM'22 test set and the AeroPath dataset, whereas the TD and BD metrics dropped for the COVID-19 test set. The top five teams also had a drop in TD and BD for the COVID-19 test set, but only marginally for the best performing methods. Due to lack of readily available trained models (i.e., easy and direct inference over new inputs), a performance comparison between the top five solutions from the ATM'22 and the AeroPath dataset could not be achieved. Be it as it may, we believe to have made up for it by using other state-of-the-art methods which have shown to provide highly qualitative segmentation results on a wider range of applications or organ of interest. The sensitivity is close to 100% for all the designs tested in the current study, indicating that very few false negative branches were segmented and hence that most of the branches in the ground truth are found. Given the very limited number of voxels constituting the smallest branches, no large impact on sensitivity can be noticed, but not all branches are properly identified as both the BD and TD are not reaching 100%.

For patients with enlarged trachea, the addition of a scaling component as part of the data augmentation during training was introduced to ensure successful segmentation regardless of size. Heavier data augmentation was also found to be important to avoid disconnection between patches in the patch-wise approach. For patients with emphysema or an enlarged esophagus, many more false positives were segmented on average. False positives in the form of isolated islands can easily be removed during the postprocessing step, as long as no connection to the main airway tree can be established. However, connected false positives are harder to assess and a more sophisticated manner would be required for successful removal. In the

case of large tumors, partly blocking the airways and occluding many branches, proper segmentation becomes increasingly more difficult within the affected areas. On the other hand, cystic fibrosis causes a widening of peripheral airways, leading to slightly easier branch detection [7]. For infections such as COVID-19, the global presence of more noise in the CT scans may render the segmentation task more burdensome.

Overall, the postprocessing step had a negligible impact on segmentation performance. Some isolated fragments could be connected together as branches, thereby slightly increasing the ratio of detected branches. Nevertheless, its added-value could be assessed visually, which increases the clinical relevance of the predicted airway tree segmentation. In an attempt to reduce the number of false positives, Wang *et al.* [16] proposed a modified 3D U-Net with feature extraction from a larger area to avoid confusion with surrounding tubular structures, quite analog to the ensembling of a full volume AGU-Net model. However, the wider range of anatomical expressions from the presence of pathology in the AeroPath dataset limits its benefits.

The proposed AeroPath dataset includes patients suffering from medium to severe lung-related pathology and features a range of abnormally distorted anatomical structures and organs. Most of the patients display large primary tumors or collapsed segments partly blocking the airways in at least one lung lobe. Subsequently, the provided airways annotations in the AeroPath dataset consist of fewer branches on average than the ATM'22 dataset. Nonetheless, it can be noted that in the ATM'22 dataset the number of annotated branches decreases from 179 in the hidden test set to 167 when only COVID-19 patients are included. A likely rationale for the decrease in number of branches is the presence of a lung-related pathology, and the more severe the pathology the less branches visible. The AeroPath dataset consists only of contrast-enhanced CT scans, whereas the ATM'22 dataset are based on non-contrast scans. All methods tested in the current study were trained on non-contrast images and still produced satisfactory predictions over the contrast-enhanced samples. In contrast-enhanced CT images, blood vessels are brighter and hence easier to identify. In addition, airways run in parallel with pulmonary arteries, as such it provides some extent of guidance which could be further leveraged by providing a pulmonary arteries mask during training. The TotalSegmentator method successfully segments pulmonary vessels in both contrast and non-contrast CT scans [19] based on traditional segmentation methods. The ability to identify vessels in contrast-enhanced CT images from models trained on non-contrast images is important knowledge. However, and as expected, the number of detected blood vessels is higher in contrast-enhanced images [19].

As part of future work directions, the gathering of more patient data will be necessary for expanding the AeroPath dataset by one order of magnitude. In addition, more specific information should be gathered about the pathology affecting each included patient, currently limited partly due to privacy and data protection policies. The witnessed low specificity in segmentation performances may be due to a greater pathology impact on neighboring anatomical structures seen in the AeroPath patients. In a few cases, the predictions from the best model provided a more complete airway tree than the training ground truth, which means that true positive airway branches are therefore labeled as false positives. To overcome such challenges, Wang *et al.* [16] introduced the adjusted tree length and branch detection rates, where segments missing from the ground truth were included. However, such small branches located deep inside the lungs, seemingly correctly segmented by the trained model, are unlikely to be within the target route for bronchoscopy. As such, given our rationale to develop a better segmentation method for assisting in planning and guidance of biopsy procedures, the proposed ground truth and trained models are appropriate for our application.

## Conclusion

The AeroPath dataset will provide the research community with a different kind of benchmark test set, for testing and refining segmentation methods on CT scans where a patient's anatomy is impacted by a lung-related pathology. This has the potential to improve automatic airway tree segmentation even further. Our proposed segmentation method may not perform better than state-of-the-art methods on healthy or COVID-19 patients, but fares better and appears more robust for airway tree segmentation in patients with strong afflictions, as featured in the AeroPath dataset.

## Author Contributions

**Conceptualization:** Karen-Helene Støverud, David Bouget, Håkon Olav Leira, Tore Amundsen, Thomas Langø, Erlend Fagertun Hofstad.

**Data curation:** Karen-Helene Støverud, David Bouget, André Pedersen, Tore Amundsen, Erlend Fagertun Hofstad.

**Formal analysis:** Karen-Helene Støverud, David Bouget, André Pedersen, Erlend Fagertun Hofstad.

**Funding acquisition:** Håkon Olav Leira, Tore Amundsen, Thomas Langø.

**Investigation:** Karen-Helene Støverud, David Bouget, Håkon Olav Leira, Tore Amundsen.

**Methodology:** Karen-Helene Støverud, David Bouget, André Pedersen, Håkon Olav Leira, Thomas Langø, Erlend Fagertun Hofstad.

**Project administration:** Håkon Olav Leira, Tore Amundsen, Thomas Langø, Erlend Fagertun Hofstad.

**Resources:** Håkon Olav Leira, Thomas Langø.

**Software:** Karen-Helene Støverud, David Bouget, André Pedersen, Håkon Olav Leira, Erlend Fagertun Hofstad.

**Supervision:** David Bouget, André Pedersen, Håkon Olav Leira, Thomas Langø, Erlend Fagertun Hofstad.

**Validation:** Karen-Helene Støverud, David Bouget, André Pedersen, Håkon Olav Leira, Erlend Fagertun Hofstad.

**Visualization:** Karen-Helene Støverud, David Bouget, Erlend Fagertun Hofstad.

**Writing – original draft:** Karen-Helene Støverud, David Bouget, André Pedersen, Thomas Langø, Erlend Fagertun Hofstad.

**Writing – review & editing:** Karen-Helene Støverud, David Bouget, André Pedersen, Håkon Olav Leira, Tore Amundsen, Thomas Langø, Erlend Fagertun Hofstad.

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
