## [Decision Letter · Decision Letter 0]

13 Aug 2024

PONE-D-24-25648AeroPath: An airway segmentation benchmark dataset with challenging pathologyPLOS ONE

Dear Dr. Langø,

Thank you for submitting your manuscript to PLOS ONE. After careful consideration, we feel that it has merit but does not fully meet PLOS ONE’s publication criteria as it currently stands. Therefore, we invite you to submit a revised version of the manuscript that addresses the points raised during the review process.

We look forward to receiving your revised manuscript.

Kind regards,

Peng Geng

Academic Editor

PLOS ONE

Journal Requirements:

2. Thank you for stating the following in the Acknowledgments Section of your manuscript: "Data were processed in digital labs at HUNT Cloud, Norwegian University of Science and Technology (NTNU), Trondheim, Norway. K.H.S., D.B., A.P., T.L., and E.F.H are partly funded by the Ministry of Health and Care Services of Norway through the Norwegian National Research Center for Minimally Invasive and Image-Guided Diagnostics and Therapy (MiDT) at St. Olavs hospital, Trondheim University Hospital, Trondheim, Norway. The research leading to these results has in addition received funding from the Norwegian Financial Mechanism 2014-2021 under the project RO-NO2019-0138, 19/2020 “Improving Cancer Diagnostics in Flexible Endoscopy using Artificial Intelligence and Medical Robotics” IDEAR, Contract No. 19/2020."

Please remove any funding-related text from the manuscript and let us know how you would like to update your Funding Statement. Currently, your Funding Statement reads as follows: "“The authors received no specific funding for this work."

Additional Editor Comments:

The author provides an airway segmentation benchmark dataset, which is an excellent contribution for this research field. In the abstract, the author mentioned "present a multiscale fusion design for automatic airway segmentation", but did not provided the specific technical implementation and structure diagram of this model. Furthermore, I suggest that the author further revise the reviewers' suggestions.

Reviewers' comments:

Reviewer's Responses to Questions

**Comments to the Author**

1. Is the manuscript technically sound, and do the data support the conclusions?

Reviewer #1: Yes

Reviewer #2: Yes

2. Has the statistical analysis been performed appropriately and rigorously? 

Reviewer #1: Yes

Reviewer #2: Yes

3. Have the authors made all data underlying the findings in their manuscript fully available?

Reviewer #1: Yes

Reviewer #2: Yes

4. Is the manuscript presented in an intelligible fashion and written in standard English?

Reviewer #1: Yes

Reviewer #2: Yes

5. Review Comments to the Author

Reviewer #1: This paper employs a robust methodology for training and validating the AGU-Net architecture using a 5-fold cross-validation approach on the ATM’22 dataset. The authors benchmark their proposed method against multiple established baseline methods, including FAST, BronchiNet, TotalSegmentator, and MEDSeg, which provides a comprehensive evaluation of performance metrics.

This paper employed a 5-fold cross-validation approach for training and validating their models, which is a standard practice in machine learning to ensure that the model's performance is robust and not overly fitted to a specific subset of the data. It also utilized multiple performance metrics, including Dice Similarity Coefficient (DSC), precision, sensitivity, and specificity, to evaluate the segmentation results against various baseline methods.

The AeroPath dataset, which includes the annotated CT scans used in their study, and the best trained model are openly accessible.

Reviewer #2: This manuscript contains two contributions to the analysis of CT images of the lungs in order to support early diagnosis and improved treatment. The first is a description of a new benchmark dataset with 27 cases of more severe pathologies than available in existing datasets. The second is a description of a multiscale fusion design for automated airway segmentation. Only the first contribution is mentioned in the title of the paper which is somewhat misleading. I suggest the title is modified to cover both parts of the paper. Also in the text of the paper the description of the dataset and the developed segmentation model is very much integrated. The paper could become more clear if this was more separated.

The paper is well written and I only have a few detailed comments and suggestions:

In the description of data augmentation it is stated that a number of transforms were applied to each input sample with a probability of 50%. But several of the transforms are not binary but rather includes a choice of rotation in the range +20 to +20 degrees, translation up to 20% of the axis and zoom in the range 50 through 150%. They can not be chosen with 50% probability. Please clarify how these augmentations were chosen. Also were all kinds of augmentations used for all samples?

It is described in the section on validation studies that the developed method could not be compared to the five best performing methods from the ATM22 challenge due to lack of code or trained models availability. In the abstract it is stated that the developed model was evaluated against competitive open source methods. It would be fair to note there the problem with access to compare against the best methods or at least to discuss this issue further.

In the evaluation of methods performance two custom defined measures are used TD and BD. The meanings of those terms are briefly described in the Metric section but the definition could be made a bit more mathematically well defined. Also in the Tables 2 and 3 the terms TD and BD (and DSC) are used without explanation, making it necessary to read the text to understand the tables. It would make the paper easier to read if the definitions were included in the table legends.

It is stated that the ground truth annotations were partly based on FAST+BronchiNet while FAST+BronchiNet is one of the methods being compared in the tables. Please comment about how or to what extent this may bias the comparisons results.

6. PLOS authors have the option to publish the peer review history of their article (what does this mean?). If published, this will include your full peer review and any attached files.

Reviewer #1: **Yes: **YIYI TAO

Reviewer #2: No

---

## [Author Response · Author response to Decision Letter 0]

30 Aug 2024

Reviewers' comments:

Reviewer #1: 

This paper employs a robust methodology for training and validating the AGU-Net architecture using a 5-fold cross-validation approach on the ATM’22 dataset. The authors benchmark their proposed method against multiple established baseline methods, including FAST, BronchiNet, TotalSegmentator, and MEDSeg, which provides a comprehensive evaluation of performance metrics.

This paper employed a 5-fold cross-validation approach for training and validating their models, which is a standard practice in machine learning to ensure that the model's performance is robust and not overly fitted to a specific subset of the data. It also utilized multiple performance metrics, including Dice Similarity Coefficient (DSC), precision, sensitivity, and specificity, to evaluate the segmentation results against various baseline methods.

The AeroPath dataset, which includes the annotated CT scans used in their study, and the best trained model are openly accessible.

Thank you for the positive response.

Reviewer #2: 

This manuscript contains two contributions to the analysis of CT images of the lungs in order to support early diagnosis and improved treatment. The first is a description of a new benchmark dataset with 27 cases of more severe pathologies than available in existing datasets. The second is a description of a multiscale fusion design for automated airway segmentation. Only the first contribution is mentioned in the title of the paper which is somewhat misleading. I suggest the title is modified to cover both parts of the paper. Also in the text of the paper the description of the dataset and the developed segmentation model is very much integrated. The paper could become more clear if this was more separated.

We agree with the comment and adjusted the title as follows:

“AeroPath: An airway segmentation benchmark dataset with challenging pathology and baseline method”

In addition, we have restructured the Materials and Methods section to further highlight our two contributions over the proposed dataset and segmentation method. Together with the newly added figure portraying the architecture design, we believe the section to be clearer to the reader.

The paper is well written and I only have a few detailed comments and suggestions:

In the description of data augmentation it is stated that a number of transforms were applied to each input sample with a probability of 50%. But several of the transforms are not binary but rather includes a choice of rotation in the range +20 to +20 degrees, translation up to 20% of the axis and zoom in the range 50 through 150%. They can not be chosen with 50% probability. Please clarify how these augmentations were chosen. Also were all kinds of augmentations used for all samples?

We have revised the phrasing in the manuscript to make it clearer to the reader. Each data augmentation method had a probability of 50% to be applied for any new input sample, in a consecutive fashion. For the randomly selected data augmentation techniques, a random value inside the described ranges was then picked.

It is described in the section on validation studies that the developed method could not be compared to the five best performing methods from the ATM22 challenge due to lack of code or trained models availability. In the abstract it is stated that the developed model was evaluated against competitive open source methods. It would be fair to note there the problem with access to compare against the best methods or at least to discuss this issue further.

The topic has been further addressed in the Discussion section. We do agree that a comparison to the ATM22 methods would have been an interesting inclusion into the article but unfortunately, we do not have any control over the accessibility of other methods. However, we believe to have used strong state-of-the-art methods to compare our method against.

In the evaluation of methods performance two custom defined measures are used TD and BD. The meanings of those terms are briefly described in the Metric section but the definition could be made a bit more mathematically well defined. Also in the Tables 2 and 3 the terms TD and BD (and DSC) are used without explanation, making it necessary to read the text to understand the tables. It would make the paper easier to read if the definitions were included in the table legends.

We have added the mathematical equation as suggested and also expanded the Table legend to make it easier to read.

It is stated that the ground truth annotations were partly based on FAST+BronchiNet while FAST+BronchiNet is one of the methods being compared in the tables. Please comment about how or to what extent this may bias the comparisons results.

We agree with the reviewer and have included text clearifying this in the Discussion section.

---

## [Decision Letter · Decision Letter 1]

13 Sep 2024

AeroPath: An airway segmentation benchmark dataset with challenging pathology and baseline method

PONE-D-24-25648R1

Dear Dr. Langø,

We’re pleased to inform you that your manuscript has been judged scientifically suitable for publication and will be formally accepted for publication once it meets all outstanding technical requirements.

Kind regards,

Peng Geng

Academic Editor

PLOS ONE

Additional Editor Comments (optional):

Reviewers' comments:

Reviewer's Responses to Questions

**Comments to the Author**

1. If the authors have adequately addressed your comments raised in a previous round of review and you feel that this manuscript is now acceptable for publication, you may indicate that here to bypass the “Comments to the Author” section, enter your conflict of interest statement in the “Confidential to Editor” section, and submit your "Accept" recommendation.

Reviewer #1: All comments have been addressed

Reviewer #2: All comments have been addressed

2. Is the manuscript technically sound, and do the data support the conclusions?

Reviewer #1: Yes

Reviewer #2: Yes

3. Has the statistical analysis been performed appropriately and rigorously? 

Reviewer #1: Yes

Reviewer #2: Yes

4. Have the authors made all data underlying the findings in their manuscript fully available?

Reviewer #1: Yes

Reviewer #2: Yes

5. Is the manuscript presented in an intelligible fashion and written in standard English?

Reviewer #1: Yes

Reviewer #2: Yes

6. Review Comments to the Author

Reviewer #1: (No Response)

Reviewer #2: My comments have been addressed, and I have no further comments

My comments have been addressed, and I have no further comments

7. PLOS authors have the option to publish the peer review history of their article (what does this mean?). If published, this will include your full peer review and any attached files.

Reviewer #1: **Yes: **Yiyi Tao

Reviewer #2: **Yes: **Ewert Bengtsson

---

## [Editor Report · Acceptance letter]

20 Sep 2024

PONE-D-24-25648R1 

PLOS ONE

Dear Dr. Langø, 

I'm pleased to inform you that your manuscript has been deemed suitable for publication in PLOS ONE. Congratulations! Your manuscript is now being handed over to our production team.

Kind regards, 

on behalf of

Dr. Peng Geng 

Academic Editor

PLOS ONE